

# 4DVarNet-SSH: end-to-end learning of variational interpolation schemes for nadir and wide-swath satellite altimetry

Maxime Beauchamp[1], Quentin Febvre[1], Hugo Georgentum[1], and Ronan Fablet[1]

[1]IMT Atlantique Bretagne Pays de la Loire, 655 Av. du Technopôle, 29280 Plouzané, France

**Correspondence:** Maxime Beauchamp (maxime.beauchamp@imt-atlantique.fr)

**Abstract.** The reconstruction of sea surface currents from satellite altimeter data is a key challenge in spatial oceanography, especially with the upcoming wide-swath SWOT (Surface Ocean and Water Topography) altimeter mission. Operational systems however generally fail to retrieve mesoscale dynamics for horizontal scales below 100km and time-scale below 10 days. Here, we address this challenge through the 4DVarnet framework, an end-to-end neural scheme backed on a variational data assimilation formulation. We introduce a parametrization of the 4DVarNet scheme dedicated to the space-time interpolation of satellite altimeter data. Within an observing system simulation experiment (NATL60), we demonstrate the relevance of the proposed approach both for nadir and nadir+swot altimeter configurations for two contrasted case-study regions in terms of upper ocean dynamics. We report relative improvement with respect to the operational optimal interpolation between 30% and 60% in terms of reconstruction error. Interestingly, for the nadir+swot altimeter configuration, we reach resolved space-time scales below 70km and 7days. The code is open-source to enable reproductibility and future collaborative developments. Beyond its applicability to large-scale domains, we also address uncertainty quantification issues and generalization properties of the proposed learning setting. We discuss further future research avenues and extensions to other ocean data assimilation and space oceanography challenges.

## 1 Introduction

Satellite altimetry is the main data source for the observation and reconstruction of sea surface dynamics on a global scale (Chelton et al., 2001). Current satellite altimeters only deliver along-track nadir observations. This results in a very scarce sampling of the ocean surface. Interpolation schemes are then key components of the operational processing of satellite altimetry data. Current operational products (Taburet et al., 2019; Lellouche et al., 2018) show however a limited ability to retrieve the full-range of mesoscale dynamics. Upcoming wide-swath altimetry SWOT mission, see e.g. (Gaultier et al., 2015), will provide for the first time two-dimensional observation of the sea surface height. The space-time sampling of satellite altimeters will however still remain scarce for a long time, which has motivated a recent research literature towards the improvement of the interpolation of satellite-derived SSH fields, see e.g. (Lopez-Radcenco et al., 2019; Lguensat et al., 2017; Beauchamp et al., 2021; Ballarotta et al., 2019).





Besides operational schemes based on optimal interpolation techniques (Taburet et al., 2019) and data assimilation schemes
for ocean circulation models (Benkiran et al., 2021), we may sort the proposed SSH interpolation schemes into three main cat-
egories: extension of optimal interpolation approaches towards multi-scale schemes (Ardhuin et al., 2020), data assimilation
schemes using sea surface dynamical priors such as quasi-geostrophic (QG) dynamics (Le Guillou et al., 2020), and data-driven
interpolation methods. The latter comprises both EOF-based (Empirical Orthogonal Function) techniques (Beckers and Rixen,

2003b; Alvera-Azcárate et al., 2009), analog approaches (Lguensat et al., 2017; Tandeo et al., 2020) and more recently deep
learning schemes (Fablet et al., 2020; Fablet and Chapron, 2022; Manucharyan et al., 2021; Beauchamp et al., 2020) which
relates to their recent application to computational imaging problems.

Here, we explore further this avenue and more specifically the 4DVarNet framework recently introduced in Fablet et al. (2021).

As it relies on a variational data assimilation formulation, it appears particularly suited to the space-time interpolation of sea
surface variables from irregularly-sampled observations. We propose a parametrization of the 4VarNet scheme dedicated to
SSH interpolation from satellite altimeter data and report OSSE (Observing System Simulation Experiment) results to support
the relevance of the proposed scheme. Our main contributions are as follows:

- The proposed 4DVarNet-SSH scheme delivers an end-to-end neural architecture using as inputs raw satellite altimeter

data and optimally-interpolated fields. We also address uncertainty quantification issues using an ensemble method.

- For OSSE on two case-study regions, respectively along the GULFSTREAM and for an open ocean area dominated by
    mesoscale eddy dynamics, the 4DVarNet-SSH scheme outperforms previous work and significantly improves perfor-
    mance metrics with respect to the operational processing. We also support the relevance of wide-swath SWOT altimeter
    data to significantly improve the reconstruction of sea surface dynamics compared to nadir-only satellite altimeters.

- We deliver an open source code for the proposed 4DVarNet-SSH scheme. It relies on a Pytorch and associated state-of-
    the-art packages. As such, it supports multi-GPU configuration and can scale up to large-scale domains.

We believe these contributions to contribute to the development of deep learning approaches for satellite altimetry, and more
broadly for operational oceanography.

This paper is organized as follows. Sect. 2 briefly reviews key methodological aspects and related work. We describe the
proposed 4DVarNet-SSH approach in Sect. 3 and Sect. 4 presents the considered OSSE setting. We report our results in Sect.
5 and discuss further our main contributions in Sect. 6.





## 2 Background and related work

From a methodological point of view, interpolation problems in geoscience are classically regarded as data assimilation issues
(Asch et al., 2016). They aim at estimating the state $\mathbf{x}_t$ of a multi-dimensional dynamical system:

$$
\begin{cases}
\frac{d\mathbf{x}_t}{dt} &= \mathscr{M}(\mathbf{x}_t) + \eta_t \\
\mathbf{y}_t &= \mathscr{H}_t(\mathbf{x}_t) + \varepsilon_t
\end{cases}.
\tag{1}
$$

The first equation relates to the forecast step which describes the evolution of the system from time $t$ to $t + dt$ according to the
potentially non-linear model $\mathscr{M}$. The second equation introduces the observations $\mathbf{y}_t$ at time $t$ where $\mathscr{H}_t$ is the corresponding
observation operator, usually known, but also potentially trainable. $\eta(t)$ is the model error and $\varepsilon(t)$ the observation error. Both
errors are generally assumed to be Gaussian, unbiased and uncorrelated over time. When discretized on a spatio-temporal grid
where index $k = 1, \cdots, T$ refers to time $t_k$, their associated covariance matrices write $\mathbf{Q}_k \in \mathbb{R}^{m \times m}$ and $\mathbf{R}_k \in \mathbb{R}^{p_k \times p_k}$.

Broadly speaking, a vast family of data assimilation methods stems from the minimization of some energy or functional which
involves two terms, a dynamical prior and an observation term. We may distinguish two main categories of data assimilation
approaches (Evensen, 2009): variational and statistical data assimilation. Specifically, within a variational data assimilation
framework, the state analysis $\mathbf{x}^a$ results in a gradient-based minimization of the defined variational cost $\mathscr{J}(\mathbf{x}) = \mathscr{J}_\Phi(\mathbf{x}, \mathbf{y}, \Omega)$
(Asch et al., 2016). The latter generally combines the sum of an observation term and a regularization term involving an
operator $\Phi$:

$$
\mathscr{J}_\Phi(\mathbf{x}, \mathbf{y}, \Omega) = \frac{1}{2}||\mathbf{y} - \mathscr{H}(\mathbf{x})||^2_{\mathbf{R}^{-1}} + \frac{1}{2}||\mathbf{x} - \Phi(\mathbf{x})||^2_{\mathbf{Q}^{-1}}
$$

$$
\tag{2}
$$

In a weak-constrained 4DVar scheme (Carrassi et al., 2018), prior operator $\Phi$ is a time-stepping operator associated with
dynamical model $\mathscr{M}$. $\mathscr{H}$ is the observation operator, $\Omega = \{\Omega_k\}$ is the set of subdomains of $\mathscr{D}$ with observations at time $t_k$,
$k = 1, \cdots, T$. $\mathbf{Q}$ and $\mathbf{R}$ are, respectively, the background and the observation error covariance matrices and $\Phi(\mathbf{x})$ denotes here
the background estimation, i.e. the physical prior, more often noted as the deterministic forecast $\mathbf{x}^b$.

Regarding statistical data assimilation, many state-of-the-art methods rely on optimal formulation (OI), Interestingly, the an-
alyzed state obtained from OI matches the minimization of the 3DVar cost function, which relates to the stationary case
of 4DVar formulation described above, see e.g. Carrassi et al. (2018). This establishes the formal link between the sta-
tistical DA frameworks and the optimal control theory used in the variational formulation. Optimal Interpolation (OI) has
been used for decades (Taburet et al., 2019) for the interpolation of along-track nadir altimeter datasets and is still used
today for the operational Marine (CMEMS) and Climate (C3S) production of the E.U. Copernicus program. It involves
a significant smoothing, solving spatial scales up to 150km. Extensions of OI schemes to multi-scale to better account for
mesoscale sea surface dynamics have recently been proposed (Ardhuin et al., 2020; Ubelmann et al., 2016). Variational DA



schemes have also been widely explored for the assimilation of satellite altimeter data in ocean general circulation models, see e.g. Ngodock et al. (2015), Benkiran et al. (2021) or Li et al. (2021). Previous works have also considered quasi-

geostrophic (QG) dynamics as an approximate and reduced-order dynamical prior for sea surface dynamics, leading to state-of-the-art performance (Ubelmann et al., 2016; Le Guillou et al., 2020). Overall, BOOST-SWOT 2020 data challenge (https://github.com/ocean-data-challenges/2020a_SSH_mapping_NATL60) provides a representative benchmarking framework to assess the performance of SSH mapping schemes for nadir-only and nadir+SWOT altimetry datasets. It further stresses the limited ability of the state-of-art to retrieve fine-scale dynamics below $1°$ and 10 days.

Whereas model-driven and optimal interpolation approaches are the state-of-the-art solutions for operational products, data-driven strategies have recently emerged as promising alternatives to improve the space-time resolution of interpolated products. We may cite among others DINEOF (Beckers and Rixen, 2003a; Alvera-Azcárate et al., 2005; Alvera-Azcárate et al., 2009) and the Analog Data Assimilation, AnDA (Lguensat et al., 2017; Tandeo et al., 2020) and the recent developments of deep learning

schemes (Barth et al., 2019; Beauchamp et al., 2020) . Beauchamp et al. (2020) have reported a benchmarking experiment, which supported the relevance of data-driven schemes compared with the operational OI product. Here, we further explore deep learning approaches, and more particularly the 4DVarNet scheme (Fablet and Chapron, 2022), which bridges variational data assimilation and deep learning. As detailed herefater, we introduce a parameterization of the 4DVarNet scheme dedicated to SSH interpolation issues and demonstrate its relevance in the context of the benchmarking settings introduced in BOOST-

SWOT 2020 data challenge.

## 3   Method

This section details the proposed learning-based framework for the interpolation of satellite altimeter data. We first briefly review 4DVarNet framework recently introduced in Fablet et al. (2021) in Sect. 3.1 and present the proposed parameterization for SSH mapping from nadir and SWOT altimeter data in Sect. 3.2. We describe the resulting Pytorch package and associated

implementation details in Sect. 3.4 and the proposed learning setting in Sect. 3.3.

### 3.1   4DVarNet framework

4DVarNet framework introduced in Fablet et al. (2021) provides a generic approach for the learning of 4DVar models and solvers. They have been shown to outperform classic 4DVar solver for toy case-studies such as Lorenz-63 and Lorenz-96 dynamics, when considering partially-observed systems. 4DVarNet framework can be regarded as an extension using trainable

gradient-based solvers of the deep learning scheme, which led to the best SSH interpolation performance in our previous work (Beauchamp et al., 2020).



From a methodological point of view, 4DVarNet framework derives an end-to-end neural architecture from an underlying variational DA formulation:

$$\mathcal{J}_\Phi(\mathbf{x}, \mathbf{y}, \Omega) = \lambda_1 ||\mathbf{y} - \mathcal{H}(\mathbf{x})||^2_\Omega + \lambda_2 ||\mathbf{x} - \Phi(\mathbf{x})||^2, \tag{3}$$

where $\lambda_{1,2}$ are predefined or tunable scalar weights and we replaced the Mahalanobis norms $||.||^{-1}_R$ and $||.||^{-1}_Q$ by a standard mean-square norm for sake of simplicity. In the regularization term, we substitute to the traditional dynamical prior $\mathcal{M}$ a neural operator $\Phi$ which convolutional architecture. Then, we can exploit the automatic differentiation tools embedded in deep learning framework to consider the following iterative gradient-based solver for the minimization of variational cost $\mathcal{J}_\Phi$ w.r.t. state $\mathbf{x}$:

$$\begin{cases} g^{(i+1)} &= LSTM\left[\alpha \cdot \nabla_\mathbf{x} \mathcal{J}_\Phi(\mathbf{x}^{(i)}, \mathbf{y}, \Omega), h(i), c(i)\right] \\ x^{(i+1)} &= x^{(i)} - \mathcal{T}\left(g^{(i+1)}\right) \end{cases}, \tag{4}$$

where $\mathcal{L}$ is a convolutional LSTM model, see e.g. (Shi et al., 2015), $\alpha$ a normalization scalar and $\mathcal{T}$ a linear mapping. This iterative rule based on a trainable LSTM operator is similar to that classically used in meta-learning schemes (Andrychowicz et al., 2016). Due to the ability of LSTM models to capture long-term dependencies, it results in a trainable gradient descent with momentum.

Overall, a 4DVarNet scheme defines a neural architecture which runs a predefined number of iterative gradient-based update (see Eq. 4). The resulting neural architecture is referred as an end-to-end architecture in the sense as it uses as inputs raw observation data $y$ and an initial guess $\mathbf{x}^{(0)}$ and as outputs the reconstructed state $\widehat{\mathbf{x}}$. Let us denote by $\Psi_{\Phi,\Gamma}(\mathbf{x}^{(0)}, \mathbf{y}, \Omega)$ the output of the 4DVarNet architecture for given priors $\Phi$ and solver $\Gamma$, see Fig. 1 and Algorithm below, the initialization $\mathbf{x}^{(0)}$ of state $\mathbf{x}$ and the observations $\mathbf{y}$ on domain $\Omega$.

Then, the joint learning of operators $\{\Phi, \Gamma\}$ is stated as the minimization of a reconstruction cost, see Sect. 3.3:

$$\arg\min_{\Phi,\Gamma} \mathcal{L}(\mathbf{x}, \mathbf{x}^\star) \text{ s.t. } \mathbf{x}^\star = \Psi_{\Phi,\Gamma}(\mathbf{x}^{(0)}, \mathbf{y}, \Omega). \tag{5}$$

In Appendix A, a parameter-free fixed point version of the solver is also given, based on the previous results of (Beauchamp et al., 2020). In addition, Beauchamp et al. (2021) have already shown how the iterative gradient-based update is more efficient that the simpler fixed-point formulation.

## 3.2  4DVarNet-SSH parameterization

The proposed 4DVarNet-SSH framework aims at exploiting and improving the mapping performance of current operational OI products. Given that OI products retrieve consistent large-scale dynamics, we rely on the following multiscale decomposition:

$$\mathbf{x} = \bar{\mathbf{x}} + d\mathbf{x} + \varepsilon \tag{6}$$

where the anomaly $d\mathbf{x}$ is seen as the difference between the true state $\mathbf{x}$ and the large-scale components $\bar{\mathbf{x}}$. Regarding the observations data, let us denote by $\mathbf{y}(\Omega) = \{\mathbf{y}_k(\Omega_k)\}$ the partial and potentially noisy altimetry observations associated with





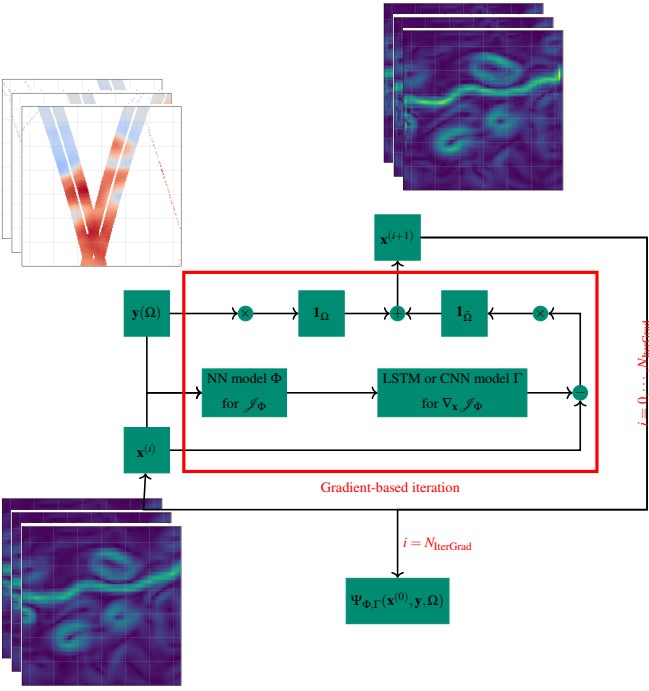

**Figure 1.** Sketch of the gradient-based algorithm: the upper-left stack of images corresponds to an example of SSH observations temporal sequence with missing data used as inputs. The upper-right stack of images is an example of intermediate reconstruction of the SSH gradient at iteration $i$ while the bottom-left stack of images identifies the updated reconstruction fields used as new inputs after each iteration of the algorithm.

---

**Algorithme 1 :** 4DVarNet algorithm, Fablet et al. (2020)

**Data :**
- $\mathbf{x} \in \mathbb{R}^{T \times m} = \{\mathbf{x}_k\}, \ k = 1, \cdots, T$
- $\mathbf{y}_\Omega = \{\mathbf{y}_{k,\Omega_k}\}, \ k = 1, \cdots, T$: observations on domains $\Omega_k \subset \mathcal{D}$
- $N_I$: number of iterations
- $\eta$: gradient step

**List of procedures :**
- Train_$\Psi_{\Phi,\Gamma}$: end-to-end learning procedure with:
  * $\Phi$: NN-based representation of the dynamical system
  * $GradLSTM$: residual NN-based representation of $\nabla_\mathbf{x} \mathcal{J}(\mathbf{x})$
  * $\Gamma$: iterative gradient-based update operator

$\mathbf{x} = \Phi(\mathbf{y})$
**while** $i < N_I$ **do**
 $\quad \mathbf{x}^{(i+1)} \leftarrow \mathbf{x}^{(i)} - \eta \times GradLSTM(\mathbf{x}^{(i)})$
 $\quad N_I \nearrow ; \ \eta \searrow ; \ i \leftarrow i + 1$
**end**
Train $\Psi_{\Phi,\Gamma}$
**Result :** $\mathbf{x}^\star \leftarrow \Psi_{\Phi,\Gamma}(\mathbf{x}^{(0)}, \mathbf{y}, \Omega)$

---





masks $\Omega = \{\Omega_k\} \subset \mathscr{D}$, where $\overline{\Omega_k}$ corresponds to the gappy part of the field and index $k$ refers to time $t_k$. We use the operational OI product as a gap-free obervation data, denoted as $\overline{\mathbf{y}}$, for state component $\overline{\mathbf{x}}$, whereas the observation data for anomaly $d\mathbf{x}$ is

$\mathbf{y} - \overline{\mathbf{y}}$ over domain $\Omega$.

Numerical experiments showed that an augmented state formulation led to better interpolation performance regarding potential stripping artifacts due to the nadir along-track sampling. This results in the application of 4DVarNet model (see Eq. 5) to augmented states $\tilde{\mathbf{x}}$ and observations $\tilde{\mathbf{y}}$ defined as follows:

$$\tilde{\mathbf{x}} = \begin{pmatrix} \overline{\mathbf{x}} \\ d\mathbf{x}_1 \\ d\mathbf{x}_2 \end{pmatrix}, \, \tilde{\mathbf{y}} = \begin{pmatrix} \overline{\mathbf{y}} \\ d\mathbf{y}_1 \\ d\mathbf{y}_2 \end{pmatrix}, \, \tilde{\Omega} = \begin{pmatrix} \mathbb{1} \\ \Omega \\ \mathbf{0} \end{pmatrix} \tag{7}$$

$$\tag{8}$$

This augmented state parameterization introduces two anomaly components. While only the first one is actually observed, the reconstructed SSH state is given by $\overline{\mathbf{x}} + d\mathbf{x}_2$.

Following Fablet et al. (2021), the operator $\Phi$ follows a purely data-driven parameterization with a two-scale residual archi-tectures involving bilinear units (Fablet et al., 2020). The number of residual blocks is set to 2 and the bilinear units are made

of two hidden convolutional layers, respectively with linear and ReLU activations, followed by a linear scheme combining the outputs of the second layer. A final convolutional layer with linear activation is involved to bring the outputs back to the initial state dimension. In its current implementation, $\Phi$ contains about 500.000 parameters. In any case, the number of gradient iterations for the solver $\Gamma$ is fixed at 5. Complementary tests showed that a higher number of iterations leads to a large increase of the training time (because of the implicit number of parameters which grows linearly with this number of iterations) without

a significant gain in terms of 4DVarnet reconstruction skills.

Regarding the initial state for iterative gradient-based rule (4), we consider the OI field $\overline{\mathbf{y}}$ for state component $\overline{\mathbf{x}}$, $\mathbf{y} - \overline{\mathbf{y}}$ for anomaly component $\mathbf{dx_1}$ and a zero state for anomaly component $\mathbf{dx_2}$. For anomaly component $\mathbf{dx_1}$, gaps are initialized to 0.

### 3.3    Learning setting

We implement a classic supervised learning strategy using gap-free targets. The considered training loss $\mathscr{L}$ combines recon-struction losses and additional regularization terms:

$$\mathscr{L}(\mathbf{x}, \mathbf{x}^\star) = \lambda_1 \sum_{i=1}^{N} \mathbf{w}_i ||\mathbf{x} - \mathbf{x}^\star||^2 + \lambda_2 \sum_{i=1}^{N} \mathbf{w}_i |\nabla_{\mathbf{x}} - \nabla_{\mathbf{x}^\star}||^2$$
$$+ \lambda_3 \sum_{i=1}^{N} \mathbf{w}_i ||\mathbf{x} - \Phi(\mathbf{x}^\star)||^2 + \lambda_4 \sum_{i=1}^{N} \mathbf{w}_i ||\mathbf{x} - \Phi(\mathbf{x})||^2 \tag{9}$$

i.e., the L2-norm of the difference between state $\mathbf{x}$ and reconstruction $\mathbf{x}^\star$ as well as for the their gradients, and regularisation

losses according to prior $\Phi$ to enforce that both the true states and the reconstructed ones are correctly encoded by prior $\Phi$. $\mathbf{w} = \{\mathbf{w}_i\}$, $i = 1, \cdots, N$ denotes a weighting vector along the data assimilation window of size $N$ (=7 here). To give more





importance to the center of the DAW, we use:

$$\mathbf{w} = \begin{bmatrix} 0 & 0.25 & 0.75 & 1 & 0.75 & 0.25 & 0 \end{bmatrix}^{\mathrm{T}} \tag{10}$$

This training loss is used in Sect. 4 for the OSSE-based BOOST-SWOT data challenge framework.

Let precise that state sequences $\mathbf{x}_{k\pm l} = \mathbf{x}_{k-l:k+l}$ of length $N = 2l + 1$ are used in the training for the interpolation problem. The idea is to optimize the results for the time $t_k$ at the center of the window $[t_{k-l}; t_{k+l}]$. The value of $N$ has to be chosen according to the dynamics of the geophysical field considered. In the following experiments, we use a value of $N = 7$ which seems to be enough to describe the spatio-temporal correlations of the anomaly between the Ground Truth and the DUACS OI scheme.

Regarding the training configuration, when the domain is small (see GULFSTREAM and OSMOSIS regions definition in Sect. 4), we use a single GPU and Adam optimizer with batch size of 2 over 200 epochs. The same set of parameters holds for larger domain (NATL and cNATL, see again Table 1) but we use the 4DVarNet-distributed version of the code over 4 GPUs. The computational time of the training procedure lies between 4 and 5 hours for the small-domain setup and between 7 and 8 hours for the large-domain setup.

## 3.4 Implementation aspects

We provide an open source PyTorch implementation of the 4dVarNet-SSH scheme[1]. Pytorch is a state-of-the-art deep learning framework. We benefit from associated packages such as lightning and hydra to provide a high-level environment and make easier the reproduction of the experiments as well as the development of other applications. Through lightning package, our implementation supports multi-GPU distributed learning configurations. This may be highly relevant to speed up the training

process.

Regarding computational issues, the OSSE-based applications, see Sect. 4, involves the processing of 7x240x240 tensors (i.e., 7-day time series over a $12° \times 12°$ domain with a $1/20°$ resolution). GPU with a significant RAM (typically above 30Go), such as NVidia V100, A40, A100, can process such tensors through the proposed 4DVarNet architecture. The direct training 4DVarNet models over larger spatial domains is however limited by the GPU memory. To address this issue, we develop a

195 specific data management module, through the so-called *dataloaders*. Our *dataloader* module automatically extracts patches of a predefined size (typically, 7x240x240 in the reported experiments) from the considered training dataset according to stride parameters as sketched in Fig.2. One can exploit the same approach to apply a learnt model to a large domain during the evaluation or production stage. In both cases, we benefit from the fully-convolutional feature of the considered neural architecture. This guarantees that, up to border effects, the 4DVarNet processing is translation-invariant.

---

[1]The code is available at https://github.com/CIA-Oceanix/4dvarnet-core



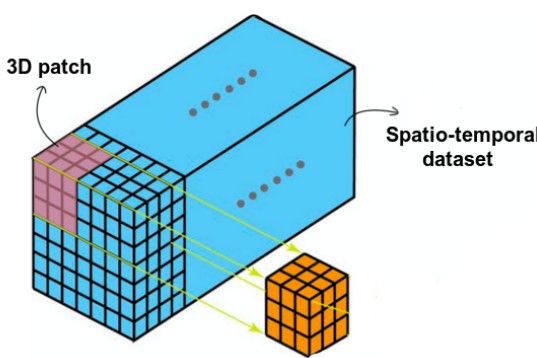

**Figure 2.** Patch-based strategy: the whole spatio-temporal dataset is split into small patches. The temporal size of the patches corresponds to the data assimilation window. The spatial size of the patches is chosen to match the maximal distance with spatial autocorrelation of the SSH

## 4 Observation System Simulation Experiments

This Section details the experimental setup considered in this study for the quantitative evaluation of the proposed framework. We first introduce the simulation dataset used in our experiments as well as the case-study regions. Sect. 4.2 reviews the simulation satellite altimetry datasets and Sect. 4.3 describes our evaluation framework.

### 4.1 NATL60 dataset and case-study regions

In our study, the Nature Run (NR) corresponds to the NATL60 configuration (Molines, 2018) of the NEMO (Nucleus for European Modeling of the Ocean) model. It is one of the most advanced state-of-the-art basin-scale high-resolution (1/60°) simulation available today, whose surface field effective resolution is about 7km.

In this work, we will use five different subdomains of the North Atlantic basin (see Fig. 3):

- two $10° \times 10°$ GULFSTREAM and GULFSTREAM2 domains,

- two $8° \times 10°$ OSMOSIS and OSMOSIS2 "open ocean" domains,

- a large $20° \times 40°$ cNATL domain, at the center North Atlantic basin, used to assess 4DVarNet training on large domains, without any pieces of land inside to avoid any issues in the learning process.

The GULFSTREAM and OSMOSIS domains (blue and red solid lines in Fig. 3) are the domains used by the BOOST-SWOT project in the framework of the NATL60 OSSE throughout the different related studies, see their 2020 ocean data challenges and Le Guillou et al. (2020). Because we aim at exploring the capabilities of 4DVarNet to deploy at the basin scale, we also propose the two alternate GULFSTREAM2 and OSMOSIS2 domains (blue and red dashed lines in Fig. 3), with similar





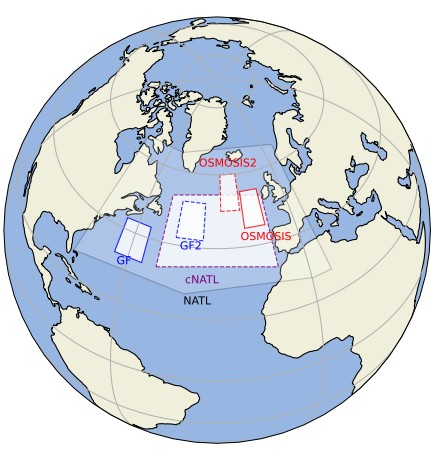

**Figure 3.** Extents of the GULFSTREAM, GULFSTREAM2, OSMOSIS, OSMOSIS2 and cNATL domains used in this work, all part of the North Atlantic (NATL) basin used in the BOOST-SWOT data challenge

dynamical properties than the two initial domains, as well as a larger domain centered in the North Atlantic basin (cNATL, purple dashed lines). The full extent of the subdomains are summarized in Table 1:

**Table 1.** Description of the NATL subdomains used for assessing 4DVarNet capabilites generalization

| Domain | longitude | latitude | extent |
|---|---|---|---|
| GULFSTREAM | [-65 °, - 55 °] | [33 °, 43 °] | 10°×10° |
| GULFSTREAM2 | [-45 °, - 35 °] | [42 °, 52 °] | 10°×10° |
| OSMOSIS | [ -19.5 °, -11.5 °] | [45 °, 55 °] | 8°×10° |
| OSMOSIS2 | [-28.5 °, -20.5 °] | [50 °, 60 °] | 8°×10° |
| cNATL | [-50 °, -10 °] | [33 °, 53 °] | 20° × 40° |
| NATL | [-79 °, 7 °] | [27 °, 65 °] | 38° × 88° |

The GULFSTREAM regions display physical processes 100 times more energetic at scales larger than 100km with a greater temporal variability than the OSMOSIS regions. As a consequence, the SSH spatial gradient at scales above 100km is lower for OSMOSIS regions which explains why we can see more small scales related structures on such domains. In addition of their intrinsic differences in terms of dynamical regimes, the latitudes of GF-based and OSMOSIS-based regions implies different SWOT temporal samplings. For OSMOSIS regions, one SWOT observation is available every day, while over the low-latitude

GULFSTREAM domains, the SWOT sampling is irregular leading to sequences of several days with only pseudo-nadir observations.

Over these regions, the Sea Surface Height (SSH) resolution of the nature run is downgraded to 1/20°, which is enough to capture both mesoscale dynamical regimes and the OSMOSIS-related smaller scales, while avoiding unnecessary heavy com-

230 putation time.



The NATL60 nature run will then be used as the reference Ground Truth (GT) in an observing system simulation experiments (OSSE). The pseudo-altimetric nadir and SWOT observational datasets will be generated by a realistic sub-sampling of satellite constellations.

## 4.2 Simulated altimetry datasets

Regarding the pseudo-nadir altimetry dataset, representative of the current pre-SWOT observational altimetric dataset, we use the groundtracks of 4 altimetric missions (TOPEX/Poseidon, Geosat, Jason-1 and Envisat) picked up from the 2003 constellation to interpolate the NATL60 simulation from October 1st, 2012 to September 29th, 2013. A Gaussian white noise with 240 variance $\sigma^2 = (4\cdots9)\mathrm{cm}^2$ is added to the interpolated NATL60 simulation by the SWOTsimulator tool to mimic a noise with a spectrum of error consistent with global estimates from the Jason-2 altimeter (Dufau et al., 2016). We aggregate the nadir pseudo-observations on a daily basis to procude the gappy daily fields used as inputs by 4DVarNet-SSH. Fig. 4(c) vs Fig. 4(d) and Fig. 5(c) vs Fig. 5(d) illustrate the resulting nadir altimetry data on 2012, October 25.

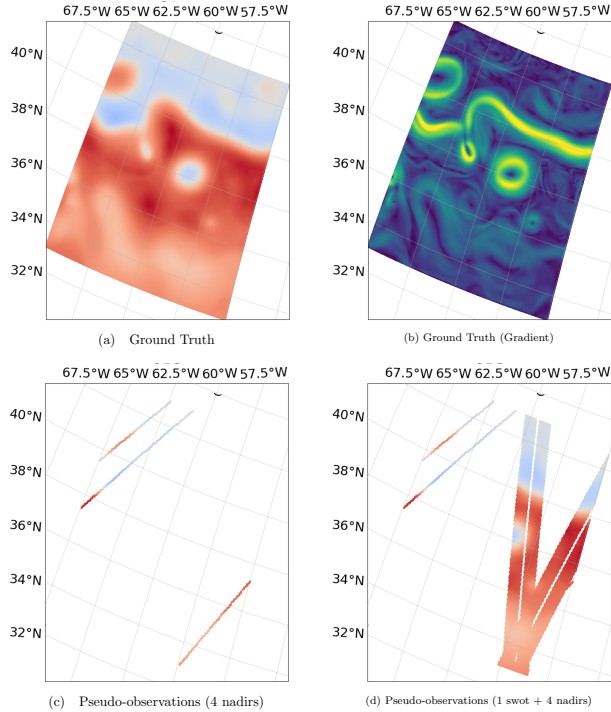

**Figure 4.** NATL60 Ground Truth (a) and its gradient (b) ; one day accumulated along-track 4 nadirs (c) and wide-swath SSH pseudo-observations + 4 nadirs (d) on 2012, October 25 (domain: GULFSTREAM)



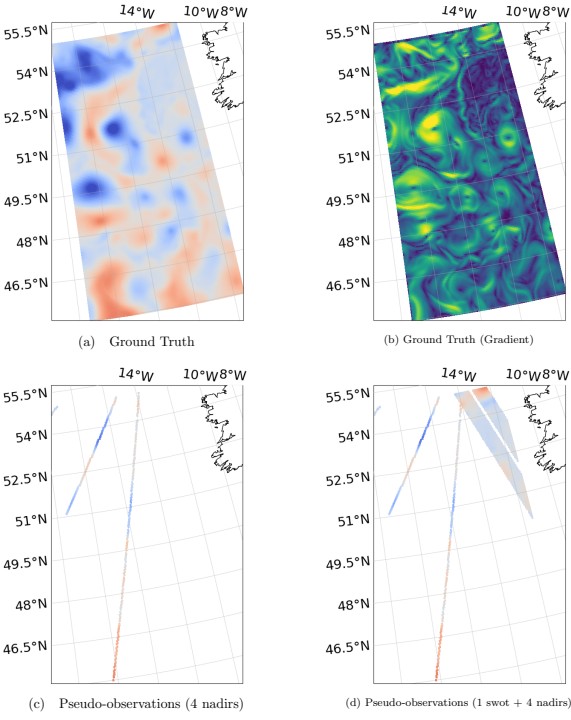

**Figure 5.** NATL60 Ground Truth (a) and its gradient (b) ; one day accumulated along-track 4 nadirs (c) and wide-swath SSH pseudo-observations + 4 nadirs (d) on 2012, October 25 (domain: OSMOSIS

We proceed similarly to simulate SWOT pseudo observations using the swotsimulator tool (Gaultier et al., 2015) in its swath mode with an along-track and across-track 2km spatial resolution (the same theoretical resolution that the upcoming SWOT mission derived products is expected to provide). Let us note that we consider error-free SWOT pseudo-observations.

### 4.3    Evaluation framework

Our evaluation framework exploits and extends the one introduced in Le Guillou et al. (2020) as follows:

**Training and evaluation setting:** We train all learning-based models using the time period from 2013, February 4 to September 30 as training period. During the training procedure, we select the best model according to metrics computed over the validation period from 2013, January 1 to February 2. Overall, we evaluate performance metrics over the test period from 2012, October 22 to December 2 for intercomparison purposes.

**Evaluation metrics:** We use BOOST-SWOT DC metrics to benchmark 4DVarnet-SSH scheme with respect to the state-of-the-art SSH interpolation schemes. They comprise: RMSE-scores, in terms of mean -$\mu$(RMSE)- and standard deviation -$\sigma$(RMSE)-, and minimal spatial and temporal scales resolved ($\lambda x$ and $\lambda t$). We refer the reader to Le Guillou et al. (2020) for the detailed description of these metrics. Besides this quantitative metrics, we analyse the space-time distribution of the interpolation error.





**Table 2.** 4DVarNet-SSH performance on the GULFSTREAM domain compared to DUACS OI (traditional covariance-based Optimal Interpolation), BFN (Back and Forth Nudging of A QG model), MIOST (Multi-scale OI), DYMOST (Dynamic OI accounting for non-linear temporal propagation of the SSH fields), and Fixed-point versions with 10 and a single iteration of the 4DVarNet solver, over the period from 2012-10-22 to 2012-12-02 (42 days)

| Method | Description | $\mu$(RMSE) | $\sigma$(RMSE) | $\lambda$x (degree) | $\lambda$t (days) |
|---|---|---|---|---|---|
| DUACS 4 nadirs | OI | 0.92 | 0.01 | 1.42 | 12.0 |
| BFN 4 nadirs | QG-based DA (nudging) | 0.92 | 0.02 | 1.23 | 10.6 |
| DYMOST 4 nadirs | Dynamic OI | 0.91 | 0.01 | 1.36 | 11.79 |
| MIOST 4 nadirs | Multi-scale OI | 0.93 | 0.01 | 1.35 | 10.19 |
| 4DVarNet 4 nadirs | Fixed-Point solver, $N_i$=10 | 0.92 | 0.01 | 1.22 | 11.51 |
| 4DVarNet 4 nadirs | NN-based 4DVar (ours) | **0.94** | **0.01** | **0.83** | **8.01** |
| DUACS 1 swot + 4 nadirs | OI | 0.92 | 0.01 | 1.22 | 11.15 |
| BFN 1 swot + 4 nadirs | QG-based DA (nudging) | 0.93 | 0.02 | 0.8 | 10.09 |
| DYMOST 1 swot + 4 nadirs | Dynamic OI | 0.93 | 0.02 | 1.2 | 10.07 |
| MIOST 1 swot + 4 nadirs | Multi-scale OI | 0.94 | 0.01 | 1.18 | 10.14 |
| 4DVarNet 1 swot + 4 nadirs | Fixed-Point solver, $N_i$=10 | 0.94 | 0.01 | 1.18 | 9.65 |
| 4DVarNet 1 swot + 4 nadirs | NN-based 4DVar (ours) | **0.95** | **0.01** | **0.62** | **5.29** |

We also explore the impact of the interpolaion onto the characterization of mesoscale eddy dynamics. Based on the work of Mason et al. (2014), we detect anticyclonic and cyclonic eddies in the Ground Truth NATL60 outputs and interpolated SSH fields using py-eddy-tracker toolbox (https://py-eddy-tracker.readthedocs.io) and analyze how key features of matching eddies, such as speed radius (km), outter radius (km), amplitude (cm) and speed max (cm/s), are retrieved.

## 5 Results

This section presents the considered OSSE for the evaluation of the 4DVarNet-SSH scheme. We first report the benchmarking experiments with respect to the state-of-the-art (Sect. 5.1). Sect. 5.2 studies the impact of wide-swath SWOT data to improve the reconstruction of finer-scale SSH pattern. Last, we analyze generalization issues and uncertainty quantitication in Sect. 5.3 and 5.4.

### 5.1 Benchmarking experiments

Regarding the BOOST-SWOT OSSE data challenge on the GULFSTREAM domain, we provide both performance with 4 nadirs and 1 swot + 4nadirs in Table 2. For both settings, the improvement is quite significant with respect to all benchmarked schemes, *i.e.* not only compared to DUACS OI (Taburet et al., 2019), but also with respect to the recently proposed SSH interpolation schemes Le Guillou et al. (2020), DYMOST (Dynamic OI accounting for the SSH non-linear temporal propagation), see e.g. Ballarotta et al. (2020) and MIOST (Multi-scale OI) of Ardhuin et al. (2020). While DUACS OI has minimal spatial and temporal resolution of 1.42°(4 nadirs)/1.22°(1 swot + 4 nadirs) and 12 days (4 nadirs)/11.15 days (1 swot + 4 nadirs), 4DVarNet-SSH reaches 0.83°(4 nadirs)/0.62°(1 swot + 4 nadirs) and 8.01 days (4 nadirs)/5.29 days (1 swot + 4 nadirs). It amounts to a gain up to **33%** in the **4 nadirs** setup and **50%** in the **1 swot + 4 nadirs** configuration.





**Table 3.** 4DVarNet-SSH performance on the OSMOSIS domain compared to DUACS OI over the period from 2012-10-22 to 2012-12-02 (42 days)

| Method | $\mu$(RMSE) | $\sigma$(RMSE) | $\lambda$x (degree) | $\lambda$t (days) |
|---|---|---|---|---|
| duacs 4 nadirs | 0.78 | 0.02 | **1.10** | 18.80 |
| 4DVarNet 4 nadirs (ours) | 0.80 | 0.01 | 1.18 | **14.51** |
| duacs 1 swot + 4 nadirs | 0.81 | 0.02 | 1.03 | 17.50 |
| 4DVarNet 1 swot + 4 nadirs (ours) | 0.87 | 0.02 | **0.35** | **6.84** |

Fig. 6 displays the SSH gradient field of DUACS OI and 4DVarNet-SSH interpolations on October, 25. The comparison to the associated Groundtruth displayed in Fig. 4(b) clearly reveals the improvement brought by 4DVarNet-SSH, in particular along the main meandrum of the GULFSTREAM.

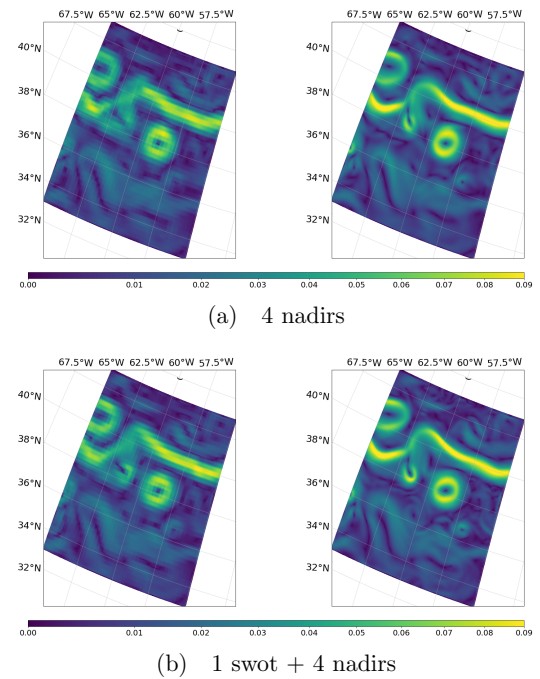

(a)    4 nadirs

(b)    1 swot + 4 nadirs

**Figure 6.** SSH Gradient (DUACS OI & 4DVarNet reconstruction) on the 2012-10-25 for the GULFSTREAM domain

We can draw similar conclusions from the experiments reported in Table 3 and Fig. 7 for the OSMOSIS domain. We may emphasize that 4DVarNet-SSH interpolation for the 1 swot + 4 nadirs configuration, see e.g. Fig.7, retrieves most of the fine-scale features of the SSH fields, which are smoothed out by the optimal interpolation.

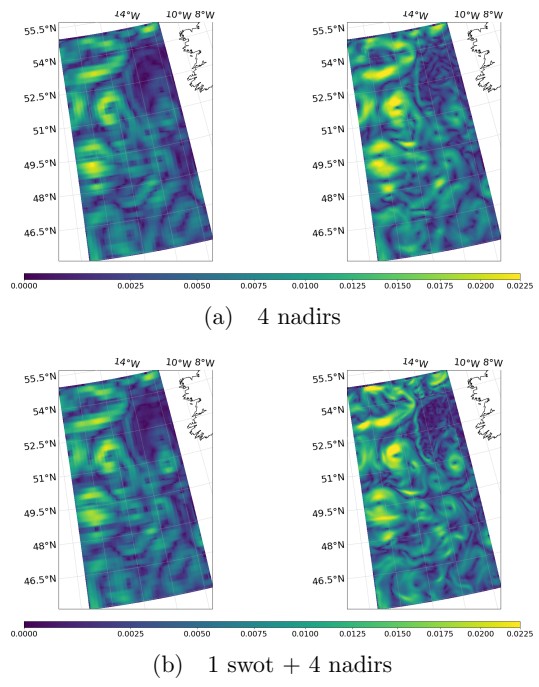

(a)  4 nadirs

(b)  1 swot + 4 nadirs

**Figure 7.** SSH Gradient (DUACS OI & 4DVarNet-SSH reconstruction) on the 2012-10-25 for the OSMOSIS domain

## 5.2  Impact of SWOT data on the interpolation performance

Thanks to its ability to reconstruct finer-scale patterns, 4DVarNet-SSH complements the assessment of the potential impact
of SWOT data onto the reconstruction of mesocale sea surface dynamics. Though the interpolation performance (Tab. 2 and
3) improves with the use of SWOT data for all the interpolation methods, the relative improvement strongly depends on the
interpolation method. Interestingly, contrary to OI DUACS scheme, we report a significant improvement when using SWOT
data with 4DVarNet-SSH for both GULFSTREAM and OSMOSIS regions. These results emphasize the ability of our scheme
to exploit irregularly-sampled high-resolution data. For instance, for the OSMOSIS region, we truly benefit from SWOT data to
reconstruct mesoscale dynamics up to $0.4°$ and 7 days, whereas OI DUACS smooths out the altimetry signals in the mesoscale
range below $1°$ and 14 days.

While we report relative gains of $20 - 25\%$ for the GULFSTREAM region for the different evaluation metrics, it reaches
$40 - 60\%$ for the OSMOSIS domain. We interpret these results as a direct consequence of differences in the space-time
sampling of SWOT data for these two regions. As revealed by Fig. 8(b) and 9(b) no SWOT data may be available over 4 (resp.
1) consecutive days for the GULFSTREAM (resp. OSMOSIS) domain, This time variability of the sampling pattern translates
for the GULFSTREAM region into a periodic variability of the MSE time series. By contrast, the OSMOSIS region leads to
a much lower time variability of the interpolation performance. The PSD-based analysis reported in Fig.8(c) and 9(c) further
supports these conclusions.



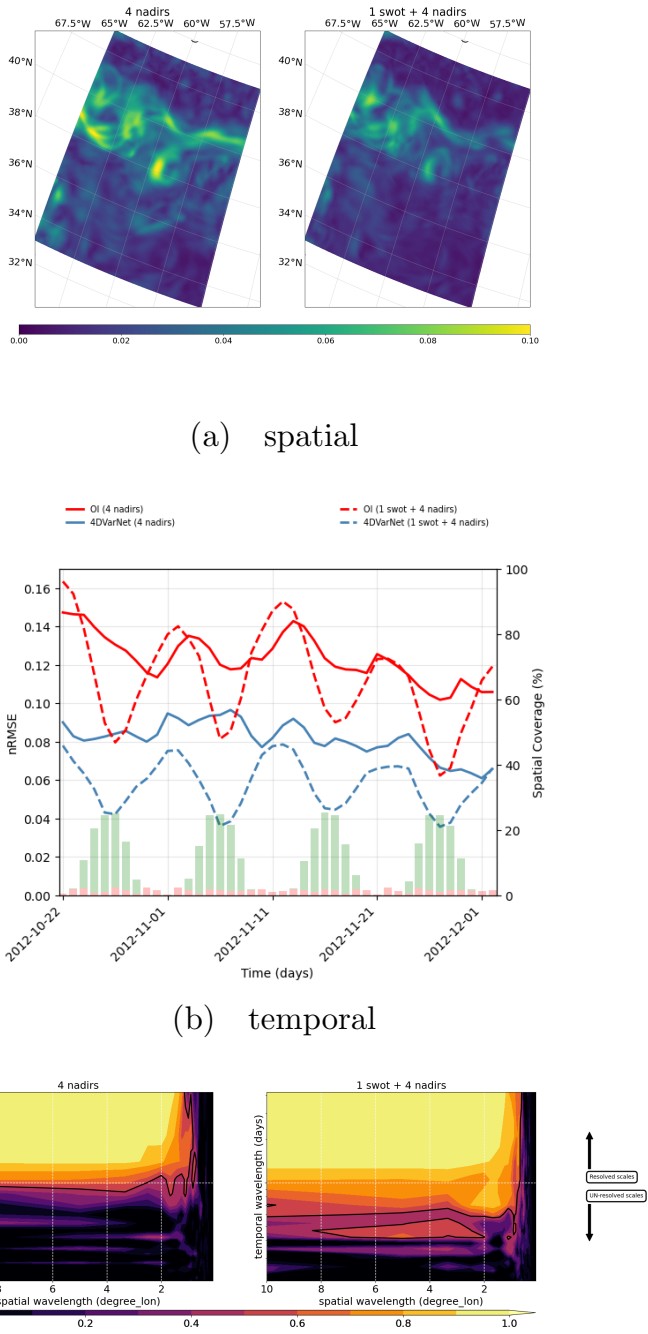

**Figure 8.** (a) Spatial performance: RMSE time series are computed for each spatial position of the GF domain (left: 4 nadirs ; right: 1 swot + 4 nadirs) ; (b) Temporal performance: RMSE daily GF maps are computed along the BOOST-SWOT DC evaluation period (left: 4 nadirs ; right: 1 swot + 4 nadirs) ; (c) Spectral performance: the PSD-based score evaluates the spatio-temporal scales resolved in GF mapping (yellow area) (top: 4 nadirs ; bottom: 1 swot + 4 nadirs)

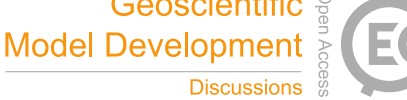

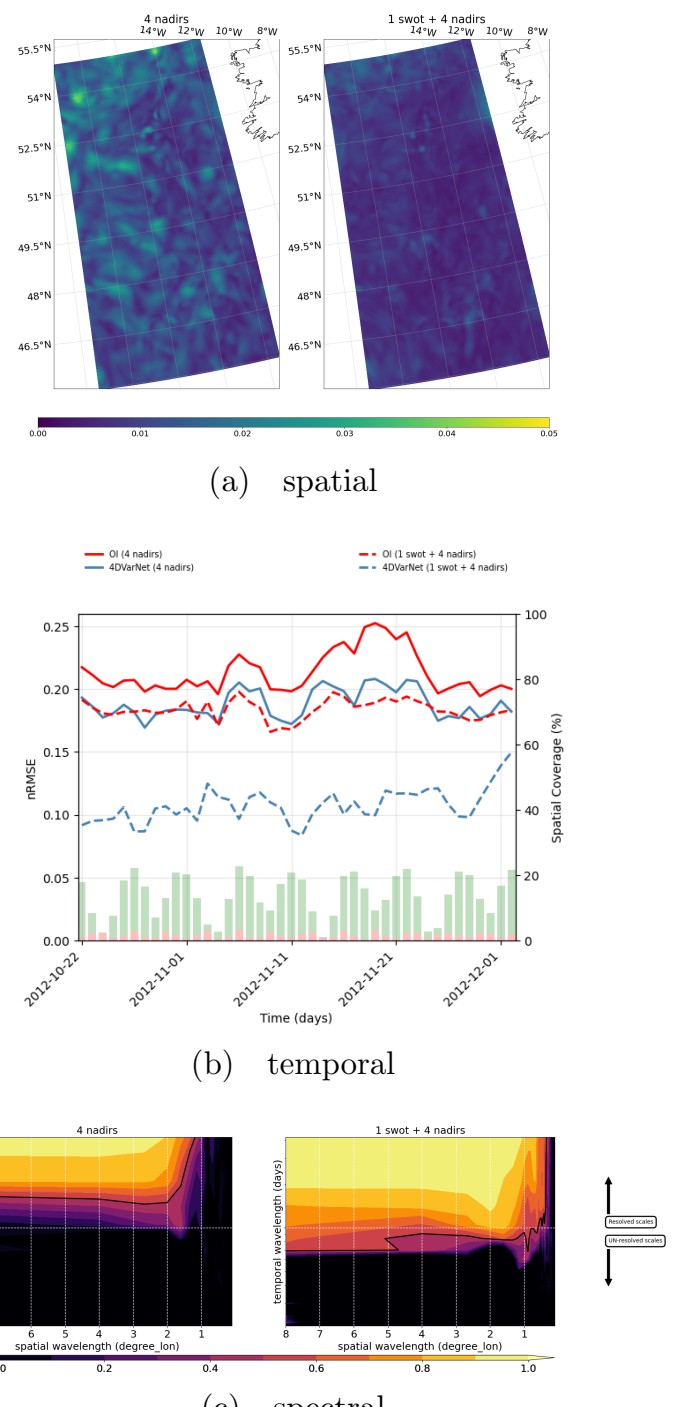

**Figure 9.** (a) Spatial performance: RMSE time series are computed for each spatial position of the GF domain (left: 4 nadirs ; right: 1 swot + 4 nadirs) ; (b) Temporal performance: RMSE daily GF maps are computed along the BOOST-SWOT DC evaluation period (left: 4 nadirs ; right: 1 swot + 4 nadirs) ; (c) Spectral performance: the PSD-based score evaluates the spatio-temporal scales resolved in GF mapping (yellow area) (top: 4 nadirs ; bottom: 1 swot + 4 nadirs)



To complement with the analysis of contribution of SWOT altimetry on the interpolation performance, Fig. 10 displays eddy
identification results on 2022, October 25 after application of a 200km high pass filter when using 1 swot + 4 nadirs configuration. Additional Figures are given in Appendix B for illustrations for both the GULFSTREAM and OSMOSIS domains in the
two observational configurations (4 nadirs and 1 swot + 4 nadirs). Clearly, 4DVarNet-SSH improves the matching between true
and interpolated eddies (39 vs 35), and the features of the matching eddies are also more similar to those of the true eddies, in
terms of speed radius (km), outter radius (km), amplitude (cm) and speed max (cm/s), with respect to their true values. Again,
the interpolation of eddy-related dynamics significantly improves with the exploitation of SWOT data.

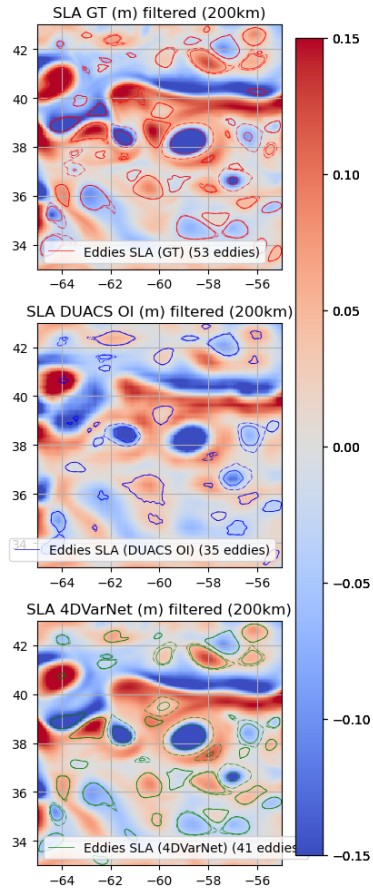

**Figure 10.** Eddies detected on the GULFSTREAM domain (2012-10-25) over SSH (1 swot + 4 nadirs)



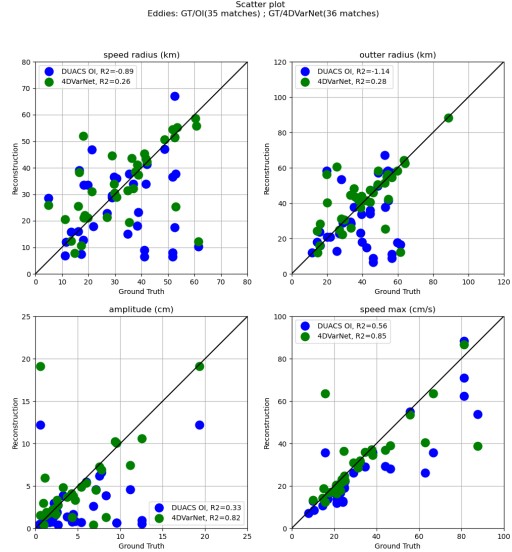

**Figure 11.** Speed radius (km), outter radius (km), amplitude (cm), speed max (cm/s) scatterplots of 4DVarNet/DUACS OI (1 swot + 4 nadirs) vs Ground truth on the GULFSTREAM domain (2012-10-25) for matching eddies

## 5.3 Generalization performance

Whereas the results reported in the previous sections involve 4DVarNet-SSH models evaluated on the same domain as the training one, we assess how 4DVarNet-SSH schemes trained for a specific domain may also apply to another one. Besides the GULFSTREAM and OSMOSIS regions, we consider three additional domains:

- cNATL domain: a larger $20° \times 40°$ North Atlantic domain, which involves a variety of dynamical regimes;

- GF2 domain: a domain similar to the reference GULFSTREAM domain in terms of upper ocean dynamics, but with a disjoint spatial extent;

- OSMOSIS2 domain: a domain similar to the reference OSMOSIS domain in terms of upper ocean dynamics, but with a disjoint spatial extent;

For the 1 swot + 4 nadirs configuration, we train 4DVarNet-SSH schemes on these three domains. We then evaluate how these models compare with the models reported in Sect. 5.1 for the GULFSTREAM and OSMOSIS domains. We also evaluate how the different models apply to the cNATL domain. Table 4 summarizes the resulting performance metrics.

As expected for each evaluation domain, we retrieve the best performance for the model trained on this domain. For the GULFSTREAM regions, the difference in terms of minimal temporal scales is negligible while the minimal spatial scales may exhibit an increase of 30% using the model trained on the GULFSTREAM2 domain. This does not hold in the other way when applying on the GULFSTREAM2 domain a model learnt on GULFSTREAM, with similar spatial scales resolved in the end.



**Table 4.** 4DVarNet performance on the GULFSTREAM and OSMOSIS domain compared to DUACS OI over the period from 2012-10-22 to 2012-12-02 (42 days)

| Domain | Method | $\mu$(RMSE) | $\sigma$(RMSE) | $\lambda$x (degree) | $\lambda$t (days) | Train/Test |
|---|---|---|---|---|---|---|
| GF | DUACS OI 1 swot + 4 nadirs | 0.92 | 0.01 | 1.22 | 11.31 | - |
| | 4DVarNet 1 swot + 4 nadirs | 0.96 | 0.01 | 0.62 | 5.29 | GF/GF |
| | 4DVarNet 1 swot + 4 nadirs | 0.95 | 0.01 | 0.86 | 5.67 | GF2/GF |
| | 4DVarNet 1 swot + 4 nadirs | 0.92 | 0.02 | 1.25 | 10.93 | cNATL/GF |
| OSMOSIS | DUACS OI 1 swot + 4 nadirs | 0.81 | 0.02 | 1.04 | 17.80 | - |
| | 4DVarNet 1 swot + 4 nadirs | 0.89 | 0.02 | 0.35 | 6.84 | OSMOSIS/OSMOSIS |
| | 4DVarNet 1 swot + 4 nadirs | 0.88 | 0.02 | 0.41 | 8.05 | OSMOSIS2/OSMOSIS |
| | 4DVarNet 1 swot + 4 nadirs | 0.84 | 0.02 | 0.93 | 9.59 | cNATL/OSMOSIS |
| cNATL | DUACS OI 1 swot + 4 nadirs | | | | | - |
| | 4DVarNet 1 swot + 4 nadirs | | | | | cNATL/cNATL |
| | 4DVarNet 1 swot + 4 nadirs | | | | | cNATL/GF |
| | 4DVarNet 1 swot + 4 nadirs | | | | | cNATL/OSMOSIS |

The same conclusions hold for the OSMOSIS regions, except that the minimal resolved temporal scales also display a slight increase (lower than 20%) over OSMOSIS. These results are consistent with the dynamical properties given in Sect. 4 and support the generalization capabilities of 4DVarNet-SSH schemes. The comparison with the performance metrics reported for the model trained on the cNATL domain suggests that the considered 4DVarNet-SSH parameterization applies to a regional scale. This training configuration only leads to a relatively marginal gain w.r.t. OI DUACS when applied to the GULFSTREAM region. We report a slightly better performance for the OSMOSIS domain. We expect future work to explore new 4DVarNet-SSH parameterizations, which could better account for basin-scale variabilities.

### 5.4 Uncertainty Quantification for 4DVarNet-SSH interpolations

**Table 5.** 4DVarNet performance on the GULFSTREAM domain based on nine different trainings with random initialization of both $\Phi$ and $\Gamma$ weights but similar training parameters (number of epochs, learning rates, optimizers, gradient steps, etc.) over the period from 2012-10-22 to 2012-12-02 (42 days)

| Members | $\mu$(RMSE) | $\sigma$(RMSE) | $\lambda$x (degree) | $\lambda$t (days) |
|---|---|---|---|---|
| 4DVarNet (#1) | 0.96 | 0.01 | 0.68 | 5.16 |
| 4DVarNet (#2) | 0.96 | 0.01 | 0.66 | 4.52 |
| 4DVarNet (#3) | 0.96 | 0.01 | 0.62 | 4.66 |
| 4DVarNet (#4) | 0.96 | 0.01 | 0.63 | 4.12 |
| 4DVarNet (#5) | 0.96 | 0.01 | 0.87 | 4.92 |
| 4DVarNet (#6) | 0.96 | 0.01 | 0.86 | 5.07 |
| 4DVarNet (#7) | 0.96 | 0.01 | 0.68 | 5.18 |
| 4DVarNet (#8) | 0.96 | 0.01 | 0.85 | 4.99 |
| 4DVarNet (#9) | 0.96 | 0.01 | 0.62 | 5.29 |
| 4DVarNet (median) | 0.96 | 0.01 | 0.67 | 4.62 |

Besides gap-free fields, operational interpolation products generally require to provide some evaluation of the reconstruction uncertainty. While this is a built-in feature of OI and statistical DA methods, uncertainty quantification may involve specific methodological or computational methods for other data assimilation schemes, among which ensemble methods represent a widely-considered family of approaches, see e.g. Asch et al. (2016). Their common feature is to generate an ensemble of solutions, generally through some randomization process.

Here, we benefit from the stochastic nature of the training procedure of 4DVarNet-SSH schemes (Goodfellow et al., 2016). Similarly to most deep learning schemes, we exploit a stochastic gradient descent during the learning stage and a random



initilization of model parameters. As such, for a given training configuration, we can learn an ensemble of 4DVarNet-SSH schemes by running multiple training procedures.

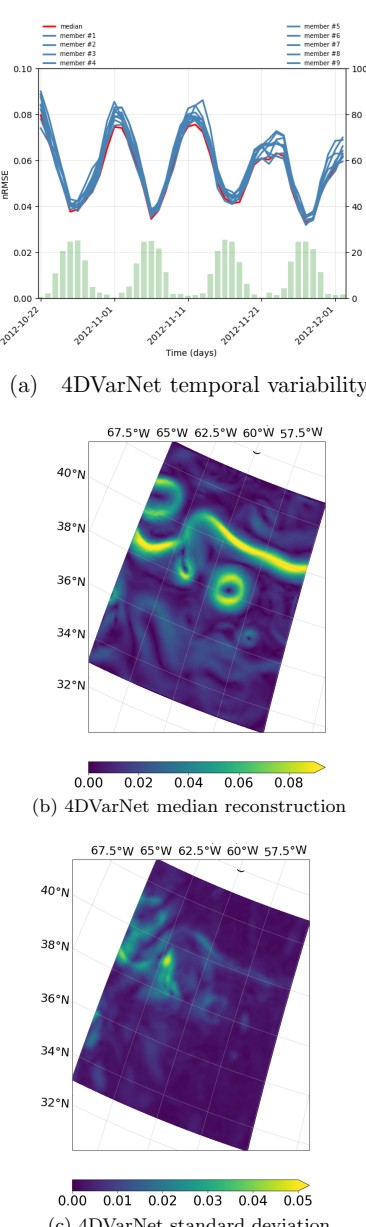

(a)    4DVarNet temporal variability

(b) 4DVarNet median reconstruction

(c) 4DVarNet standard deviation

**Figure 12.** Interpolation performance of an ensemble of nine 4DVarNet-SSH models trained using similar training parameters (number of epochs, learning rates, optimizers, gradient steps, etc.) but different random initialization of both $\Phi$ and $\Gamma$ weights. (a) : spatial RMSE time series on the BOOST-SWOT DC evaluation period ; (b) 4DVarNet median run (2012-10-25, GF domain) and (c) its spatial standard deviation



We apply this approach to build an ensemble of nine 4DVarNet-SSH schemes for a given training configuration, which comprises a training dataset, the considered 4DVarNet-SSH parameterization and given training hyperparameters (*i.e.*, number of epochs, learning rates, optimizers). For a given observation time window, we then retrieve nine interpolations, from which we can compute a median field and the associated standard deviation. We report in Tab. 5 the performance metrics for the GULF-STREAM domain of the nine trained models as well as the median model. It reveals the internal variability of the training

process. Though it does not reach the best performance, the median model combines a resolved spatial scale below 0.7° and a resolved time scale below 5 days, which is only the case of 6 over 9 of the trained models. Fig. 12(a) further illustrates this aspect. Interestingly, the standard deviation of the ensemble of 4DVarNet-SSH schemes correlates to the interpolation error, with an R2 coefficient of determination equals to 0.86, see Fig. 13. As such, it can be regarded as an indicator of the interpolation error.

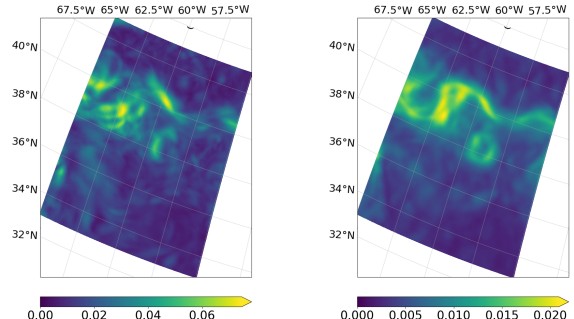

**Figure 13.** Standard deviation of the 4DVarNet-SSH median ensemble interpolation error (left) and average of the daily standard deviation interpolation errors on the test period (right)

## 350  6  Conclusion and Discussion

This paper introduced 4DVarNet-SSH scheme, an end-to-end neural architecture for the space-time interpolation of SSH fields from nadir and wide-swath satellite altimetry data. 4DVarNet-SSH scheme draws from recent methodological development to bridge data assimilation and deep learning with a view to learning 4DVar DA models and solvers from data. Numerical experiments within an OSSE setting support the relevance of 4DVarNet-SSH scheme with respect to the state-of-the-art.

We discuss further our main contributions according to three aspects: the added value of deep learning scheme for satellite altimetry and operational oceanography, the exploitation of upcoming SWOT data and the ability to scale up learning approaches from regional case-studies to the global scale.

**Deep learning for satellite altimetry and operational oceanography:** This study contributes to a growing research effort regarding the potential benefit of deep learning schemes for space and operational oceanography challenges, see e.g. Ballarotta et al. (2020). Given the sampling of available satellite and in situ data sources, interpolation problems naturally arise





as critical challenges. This study brings an additional evidence of the potential of deep learning schemes to outperform the state-of-the-art operational techniques, generally based on optimal interpolation and data assimilation. Importantly, we do not
rely on the off-the-shelf application of some reference deep learning architectures. The considered class of neural architectures relates to a variational DA formulation, such that it can be regarded as the implementation of a neural and trainable version of a DA model and solver. Our results for satellite altimetry are in line with other recent studies for other ocean parameters, such as sea surface temperature (Barth et al., 2019), suspended sediments (Vient et al., 2022) and 3D temperature and salinity fields (Pauthenet et al., 2022). All these studies support the potential of neural approaches to retrieve finer-scale
variabilities from available satellite and/or in situ observations. Regarding satellite altimetry, future challenge includes the application to real altimetry datasets, see e.g. the 2021 Observation System Experiment (OSE) BOOST-SWOT data challenge https://github.com/ocean-data-challenges/2021a_SSH_mapping_OSE, as well as the exploitation of multimodal synergies (Fablet and Chapron, 2022).

**Making the most of SWOT data:** Our study brings new evidence that the wide-swath space-time sampling of upcoming SWOT mission could lead to a very significant improvement of the reconstruction of mesoscale sea surface dynamics. For the considered case-study regions, with contrasted dynamical regimes in play and revisit times of SWOT orbits, we report relative gains from 20% to 60% compared to nadir altimetry data only in terms of RMSE and resolved space-time scales. These results assume an error-free SWOT product. Therefore, exploring further how these results could generalize to error-prone
(Esteban-Fernandez, 2014; Gaultier and Ubelmann, 2010) and uncalibrated SWOT data (Febvre et al., 2022) is a critical challenge. Preliminary preprocessing of the pseudo-SWOT observations (Metref et al., 2020) to filter out its correlated components and avoid major issues in the assimilation and/or learning process of the interpolation methods may also be considered. The extension of the considered OSSE to multi-swot configurations could also provide new means to optimize the deployment of multi-satellite configurations in coming years.

**Scaling up to a global scale with learning-based scheme:** Our numerical experiments focused mainly on a regional- scale, typically $10° \times 10°$ domains as illustrated by the GULFSTREAM and OSMOSIS regions. The reported results support the relevance of the proposed 4DVarNet-SSH parameterization to account for such regional space-time variabilities. Scaling up to a basin scale or even the global scale naturally arises as a key challenge for future work. Through the built-in features of
390 Pytorch framework and associated packages, our open-source code can leverage multi-GPU distribution learning schemes and on-the-fly mini-batch generation tools to deal with larger-scale dataset from a computational point of view. To account for a greater diversity of dynamical regimes in play on the global scale, or even on a basin scale, it also seems necessary to explore more complex 4DVarNet-SSH parameterizations, especially regarding dynamical prior $\Phi$. This could benefit from the variety of neural architectures recently introduced in computational imaging (Barbastathis et al., 2019), especially using attention
mechanisms (Vaswani et al., 2017) to achieve some decomposition of the underlying space-time variabilities.

*Code and data availability.* The open-source 4DVarNet version of the code is available on GitHub (https://github.com/CIA-Oceanix/4dvarnet-core). The datasets is shared through the BOOST-SWOT data challenge also available on Github (https://github.com/ocean-data-challenges/2020a_SSH_mapping_NATL60)

*Video supplement.* The animations corresponding to the 4DVarNet comparison to DUACS OI on the BOOST-SWOT DC test period are
400 given for both GULFSTREAM and OSMOSIS domains in the 4 nadirs and 1 swot + 4 nadirs configuration. They can be found on the AI CHair OceaniX Youtube channel:

- GF (4 nadirs): https://youtube.com/shorts/QKXukB_Rd5E

- GF (1 swot + 4 nadirs): https://youtube.com/shorts/i91Z1pMm4gY

- OSMOSIS (4 nadirs): https://youtube.com/shorts/Pxcsd0Afco0

- OSMOSIS (1 swot + 4 nadirs): https://youtube.com/shorts/HbVSJFtdG6Q

*Author contributions.* Maxime Beauchamp designed the experiments, ran the analysis of the results and wrote the paper. Ronan Fablet is the principal investigator of the 4DVarNet methodology. Quentin Febvre led the developments of the 4DVarNet implementation on large domains. Hugo Georgenthum ran the experiments used in this paper. All the authors actively participate to the open-source 4DVarNet version of the code available on GitHub (https://github.com/CIA-Oceanix/4dvarnet-core).

*Competing interests.* No competing interests are present.

*Acknowledgements.* This work was supported by LEFE program (LEFE MANU project IA-OAC), CNES (grant OSTST DUACS-HR) and ANR Projects Melody and OceaniX. It benefited from HPC and GPU resources from Azure (Microsoft Azure grant) and from GENCI-IDRIS (Grant 2020-101030).



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



## Appendix A: Fixed-point formulation of the solver

Let note that when replacing both CNN/LSTM cell by the identity operator and the minimization function $\mathscr{J}_\Phi(\mathbf{x},\mathbf{y},\Omega)$ by its single regularization term $\mathscr{J}_\Phi^b(\mathbf{x})$, the gradient-based solver simply leads to a parameter-free fixed-point version of the algorithm, the same used in Beauchamp et al. (2020); Fablet et al. (2019), which is similar to the DINEOF approach, see Fig.

A1.

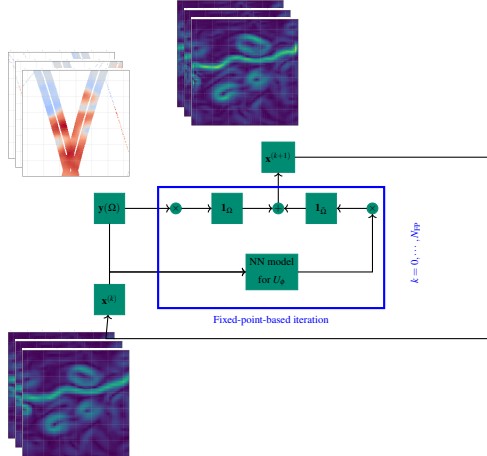

**Figure A1.** Sketch of the iterative fixed-point algorithm: the upper-left stack of images corresponds to an example of SSH observations temporal sequence with missing data used as inputs. The upper-right stack of images is an example of intermediate reconstruction of the SSH gradient at iteration $i$ while the bottom-left stack of images identifies the updated reconstruction fields used as new inputs after each iteration of the algorithm.

$$
\begin{cases}
\mathbf{x}^{(k+1)} & = \psi\left(\mathbf{x}^{(k)}\right) \\
\mathbf{x}^{(k+1)}\left(\Omega\right) & = \mathbf{y}\left(\Omega\right) \\
\mathbf{x}^{(k+1)}\left(\overline{\Omega}\right) & = \mathbf{x}^{(k+1)}\left(\overline{\Omega}\right)
\end{cases}
\tag{A1}
$$

This fixed-point solver is parameter-free and easily implemented as a neural network in a joint solution with the NN-parametrization of $\mathscr{J}_\Phi$ for the interpolation problem.

## Appendix B: Additional results on the 4DVarNet generalization capabilities





**Table B1.** 4DVarNet performance on the GULFSTREAM2 and OSMOSIS2 domain compared to DUACS OI over the period from 2012-10-22 to 2012-12-02 (42 days)

| Domain | Method | $\mu$(RMSE) | $\sigma$(RMSE) | $\lambda$x (degree) | $\lambda$t (days) | Train/Test |
|--------|--------|-------------|----------------|---------------------|-------------------|------------|
| GF2 | DUACS OI 1 swot + 4 nadirs | 0.87 | 0.02 | 1.26 | 11.95 | - |
| | 4DVarNet 1 swot + 4 nadirs | 0.93 | 0.02 | 0.63 | 6.30 | GF2/GF2 |
| | 4DVarNet 1 swot + 4 nadirs | 0.92 | 0.01 | 0.66 | 6.53 | GF/GF2 |
| | 4DVarNet 1 swot + 4 nadirs | 0.88 | 0.02 | 1.26 | 11.22 | cNATL/GF2 |
| OSMOSIS2 | DUACS OI 1 swot + 4 nadirs | 0.91 | 0.02 | 1.35 | 17.69 | - |
| | 4DVarNet 1 swot + 4 nadirs | 0.96 | 0.01 | 0.70 | 33.61 | OSMOSIS2/OSMOSIS2 |
| | 4DVarNet 1 swot + 4 nadirs | 0.95 | 0.01 | 0.69 | 36.84 | OSMOSIS/OSMOSIS2 |
| | 4DVarNet 1 swot + 4 nadirs | 0.93 | 0.02 | 1.10 | 9.64 | cNATL/OSMOSIS2 |

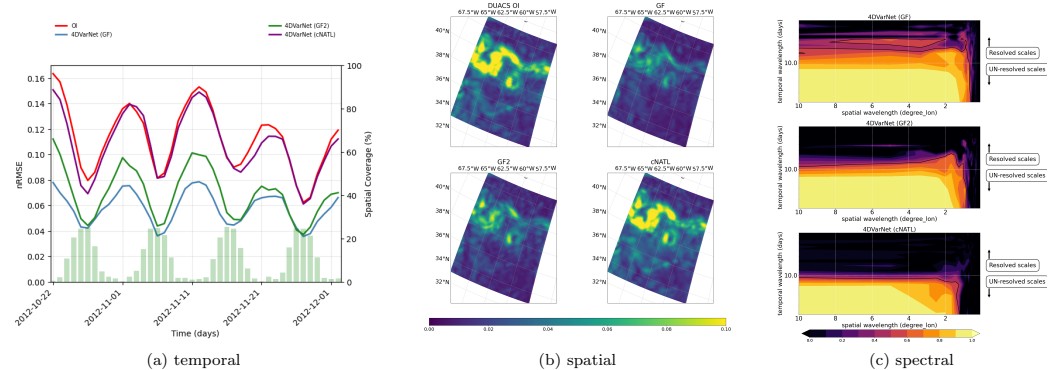

(a) temporal     (b) spatial     (c) spectral

**Figure B1.** 4DVarNet generalization capabilities (GULFSTREAM): spatial, temporal and spectral performance on the BOOST-SWOT DC evaluation period based on three different training domains: GULFSTREAM, GULFSTREAM2 and cNATL

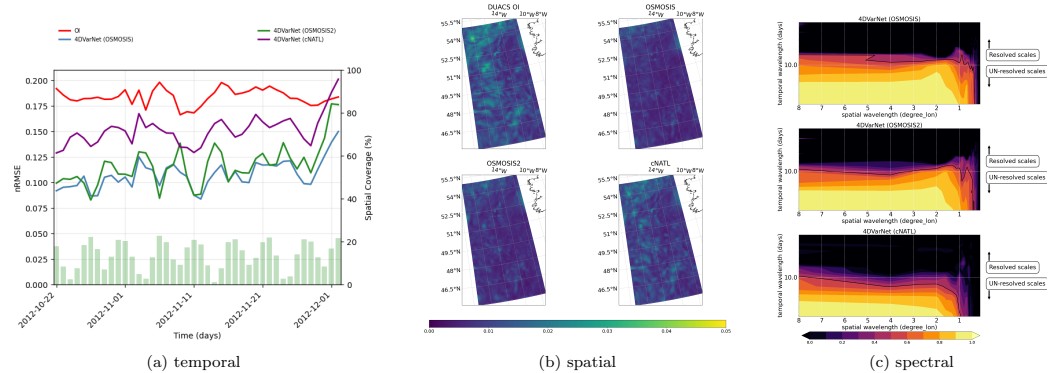

(a) temporal     (b) spatial     (c) spectral

**Figure B2.** 4DVarNet generalization capabilities (OSMOSIS): spatial, temporal and spectral performance on the BOOST-SWOT DC evaluation period based on three different training domains: OSMOSIS, OSMOSIS2 and cNATL



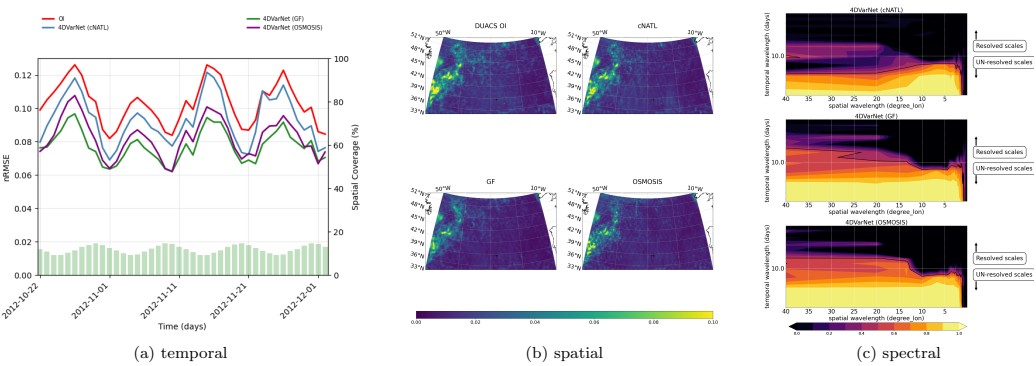

**Figure B3.** 4DVarNet generalization capabilities (cNATL): spatial, temporal and spectral performance on the BOOST-SWOT DC evaluation period based on three different training domains: cNATL, GF and OSMOSIS

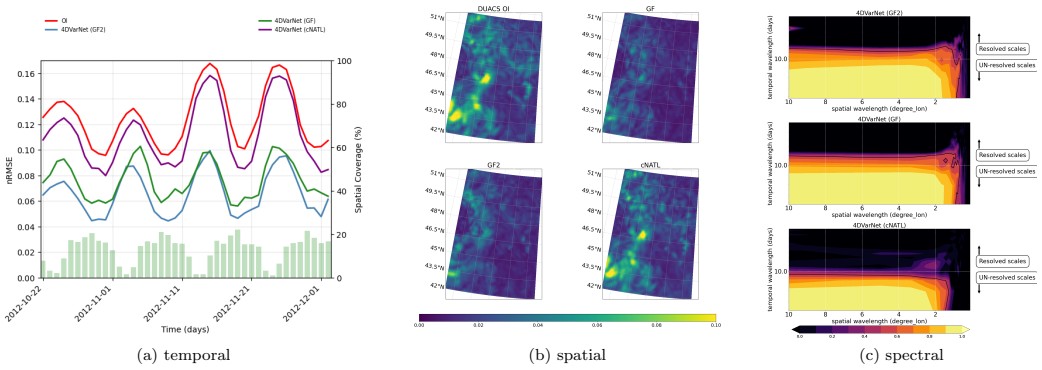

**Figure B4.** 4DVarNet generalization capabilities (GULFSTREAM2): spatial, temporal and spectral performance on the BOOST-SWOT DC evaluation period based on three different training domains: GULFSTREAM2, GULFSTREAM and cNATL

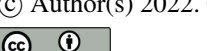



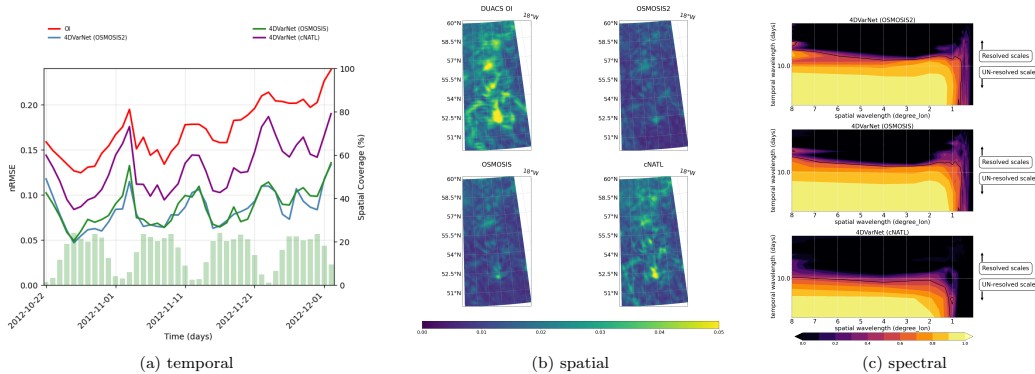

**Figure B5.** 4DVarNet generalization capabilities (OSMOSIS2): spatial, temporal and spectral performance on the BOOST-SWOT DC evaluation period based on three different training domains: OSMOSIS2, OSMOSIS and cNATL





530 **Appendix C: Eddy identifications**

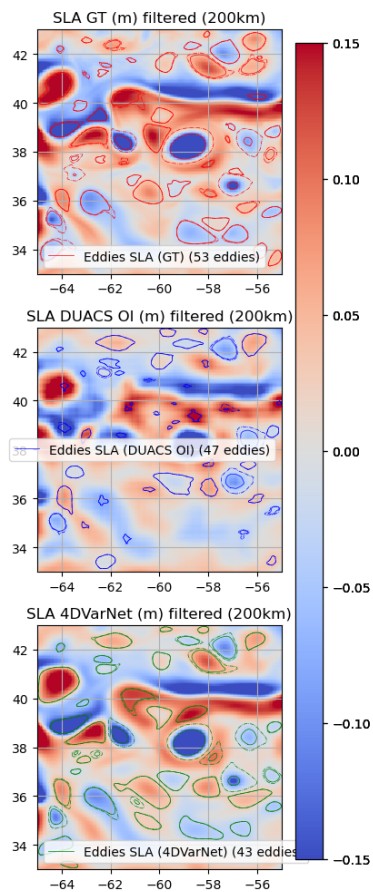

**Figure C1.** Eddies detected on the GULFSTREAM domain (2012-10-25) over SSH (4 nadirs)



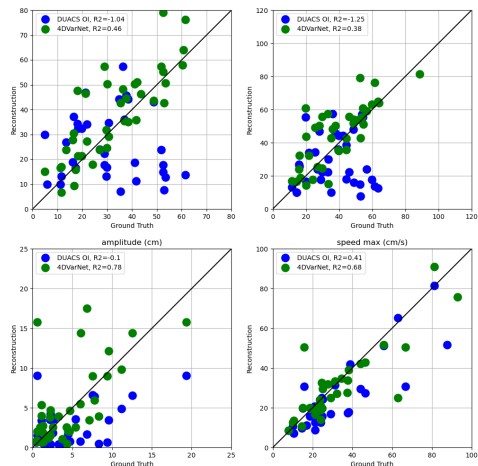

**Figure C2.** Speed radius (km), outter radius (km), amplitude (cm), speed max (cm/s) scatterplots of 4DVarNet/DUACS OI (4 nadirs) vs Ground truth on the GULFSTREAM domain (2012-10-25) for matching eddies





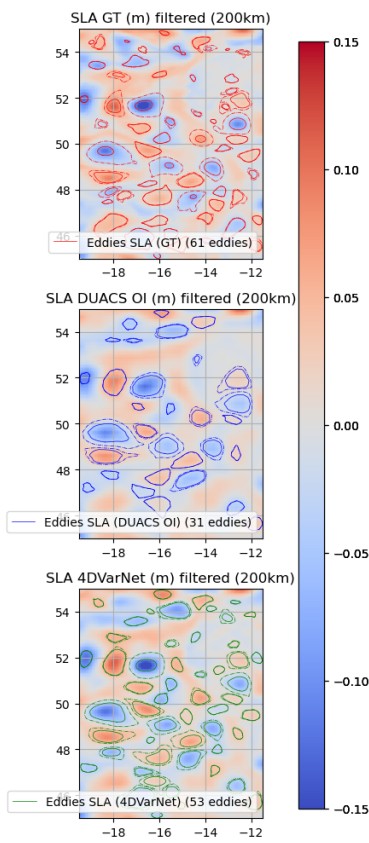

**Figure C3.** Eddies detected on the OSMOSIS domain (2012-10-25) over SSH (4 nadirs)





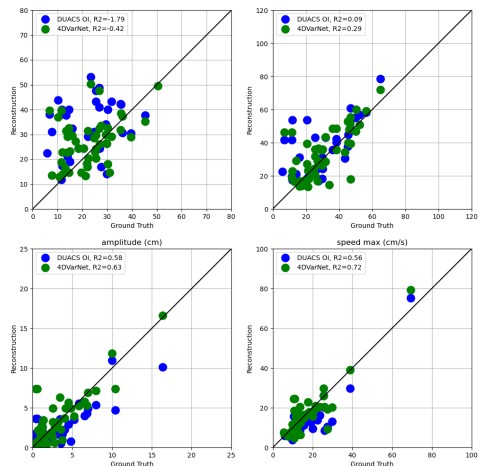

**Figure C4.** Speed radius (km), outter radius (km), amplitude (cm), speed max (cm/s) scatterplots of 4DVarNet/DUACS OI (4 nadirs) vs Ground truth on the OSMOSIS domain (2012-10-25) for matching eddies



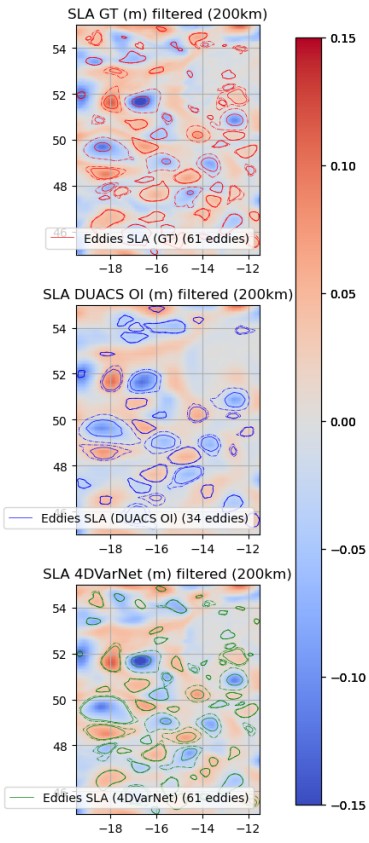

**Figure C5.** Eddies detected on the OSMOSIS domain (2012-10-25) over SSH (1 swot + 4 nadirs)



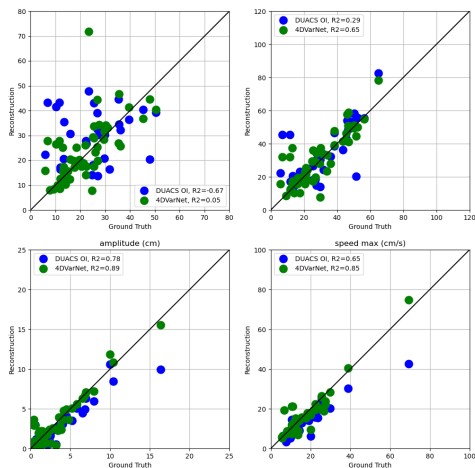

**Figure C6.** Speed radius (km), outter radius (km), amplitude (cm), speed max (cm/s) scatterplots of 4DVarNet/DUACS OI (1 swot + 4 nadirs) vs Ground truth on the OSMOSIS domain (2012-10-25) for matching eddies