# Peer review of "4DVarNet-SSH: end-to-end learning of variational interpolation schemes for nadir and wide-swath satellite altimetry"

_Geoscientific Model Development, 2022_

## Editor Comment (EC1)

This paper presents an application of the 4DVarNet framework to SSH interpolation from nadir and wide-swath satellite altimetry. While the methodology and the results are clearly of interest, the paper can not be considered for publication under its present form. In particular, the explanation of the methodology is very difficult to follow because several of the mathematical quantities involved are not even defined. The quality of the english can be largely improved. Several typos are also present. Comments are detailed below:

**Main Comments**

- The background definition is not clearly put. It begins with "the state analysis $x^a$s results in (from) a gradient-based minimization of the defined variational cost $J(x) = J_\phi(x, y, \Omega)$. None of these mathematical quantities is properly defined. $\Phi$ is said to be "a time-stepping operator associated with the dynamical model", which is far too vaue. It can be understood that $x$ and $y$ are temporal vectors of respectively the state of the system and the observations but this should be stated explicitly. Please be more rigorous in the description of the objectives at hand and the mathematical objects and tools you use.

- The paragraph linking the 4DVar formulation with the 3DVar and optimal interpolation is confusing. You should develop to make it clearer.

- Equation (4) that is supposed to be at the core of the 4DVarNet approach is not understandable under this form. The quantities $g, h$ and $c$ are not even defined. The reader can guess that $\mathcal{L}$ is the "LSTM" in the equation but otherwise, the explanation is not clear. You should take the time to explain properly what is done at each iteration of the algorithm while defining all the mathematical quantities involved. In the following paragraph, a solver $\Gamma$ comes out of nowhere. How is it linked with equation (4)? The quantities of interest that seem to be the NN $\Phi$ and $\Gamma$ are learned by minimizing a cost function that is given 3 pages later. The augmented framework that is described in section

3.2 involves two anomalies $dx_1$ and $dx_2$, neither of which are defined. Consequently, I think that section 3 should be completely rewritten to make it much more pedagogical as it is far from being understandable in its current form.

- Section 3.4 is also difficult to follow, especially how the patch are built. In can be read in the caption that "The spatial size of the patches is chosen to match the maximal distance with spatial autocorrelation of the SSH". Please develop and justify properly.

- Section 4.2 seems to explain how to build the training and test set but this should be stated explicitly.

- In the beginning of section 4.3, the training and evaluation settings are described. It is not clear to me why the test set is built upon older data with respect to the training set. Please justify.

- Concerning the evaluation metrics, the models obtaining lower values for $\lambda_t$ and $\lambda_x$ seem to be favored. You should report what these criteria are instead of just giving a reference. Also, it is not clear what the $\sigma$ is: as the approach provides only a state estimate given the observations, I do not know where this $\sigma$ comes from.

- Section 5.4 proposes a way to estimate uncertainties from several learnings of the 4DVarNet. It may be seen as a bootstrap procedure and therefore provides an estimate of the uncertainty of the prediction offered by 4DVarNet but not of the phenomenon itself. You should make a clear distinction between both.

**Minor comments**

There are lots of typos. Here I list those I have found:

- in the abstract: "SWOT (Surface Ocean and Water Topography)" reads as SOWT

- p.1 l.22: SSH is not defined

- p.2: the last sentence of the first paragraph is not understandable

- p.2 l.36: "4DVarNet" instead of "4VarNet"

- p.2 l.47: "We believe these contributions to contribute...", please rephrase

- p.2 l.59: the observation operator is "potentially trainable". Trainable means that a parametric representation of this operator exists whose parameters can be estimated. Otherwise, you should use "learnable"

- p.2 l.75: "optimal formulation (OI)" -> interpolation ?

- p.2 l.78: "framework", without the "s"

- p.2 l.81: "smoothing" instead of "smooting"

- p.5 l.118: "with" instead of "which"

- p.5 l.119: "libraries" rather than "framework"

- p.7 l.169: "...the their gradients..."

- p.13 l.259: "interpolaion"

---

## Author Response (AR1)

**Editor:**

Dear authors,
Unfortunately, after checking your manuscript, it has come to our attention that it does not comply with our "Code and Data Policy".
https://www.geoscientific-model-development.net/policies/code_and_data_policy.html
You have archived your code and data on GitHub. However, GitHub is not a suitable repository. GitHub itself instructs authors to use other alternatives for long-term archival and publishing, such as Zenodo. Therefore, please, publish your code in one of the appropriate repositories according to our policy, and reply to this comment with the relevant information (link and DOI) as soon as possible, as it should be available for the Discussions stage. Also, please, include the relevant primary input/output data. In this way, you must include in a potentially reviewed version of your manuscript the modified 'Code and Data Availability' section, the DOI of the code and another DOI for the dataset.

Please, be aware that failing to comply promptly with this request could result in rejecting your manuscript for publication.

Juan A. Añel
Geosci. Model Dev. Exec. Editor

**Response to editor:**

Dear Sir,

I think there was an issue when publishing the preprint because I already had this comment when submitting the paper and I made the appropriate changes.  But maybe the preprint was not updated. The code is available via Zenodo at:
https://doi.org/10.5281/zenodo.7186322. The final version will of course include this link instead of the Github one initially given.

Kind regards,

Maxime Beauchamp

**Reviewer #1:**

The authors present an Observing System Simulation Experiment using errorless, simulated SWOT data and a 4DVAR, machine-learning hybrid data assimilation approach. They train on the nature run from one of their experimental domains, apply the 4DVAR-Net to the same experimental domain, and then also apply the learned 4DVAR-Net to different experimental domains. The approach has merit and the algorithm shows skill in reproducing the nature run of each domain. I recommend the manuscript be published after addressing several minor comments I ran across when reviewing.

1. Line 36 - 4Var-Net - incomplete acronym

2. Line 75 - Period at end of first sentence.

3. Line 108 - How can the approximation outperform the full solver? Maybe on computational time?

4. Line 118 - ...with convolutional architecture.

5. Line 259 - interpolation

6. Figure 6 - These results are suspiously good and also underwhemling. How can the 4 nadir assimilation be doing so well and how is SWOT adding so little? Is the OSSE skill satured because it is too low resolution?

**Response to reviewer #1:**

Dear Sir or Madam,

Thanks for your comments, I will try to answer your questions:

1. Line 36 - 4Var-Net - incomplete acronym

Correction made in the new version

2. Line 75 - Period at end of first sentence.

Correction made in the new version

3. Line 108 - How can the approximation outperform the full solver? Maybe on computational time?

Yes but not only, we outperform the 4DVar solver because the latter is only optimal for Gaussian statistics and linear dynamics. Then, even if the cost function involved in the solver relies on the 4DVar cost function, the training of the joint architecture is based on the MSE w.r.t the true state (known because we are in a supervised setting). This explains why there is room for learning-based methods to improve traditional DA for non gaussian/linear inverse problems.

4. Line 118 - ...with convolutional architecture.

Correction made in the new version

5. Line 259 - interpolation

Correction made in the new version

6. Figure 6 - These results are suspiously good and also underwhemling. How can the 4 nadir assimilation be doing so well and how is SWOT adding so little? Is the OSSE skill satured because it is too low resolution?

This is not the case maybe the Figure and related text are not clear. Figure 6 has to be seen together with Table 3. If using 4 nadirs improves DUACS OI, SWOT brings a lot of improvement, both in terms of RMSE and spatio-temporal scales resolved (see again Table 3).

Please do not hesitate if you have any further questions.

Kind regards,

Maxime Beauchamp

**Reviewer #2 :**

The authors demonstrate an innovative data assimilation scheme that emulates traditional variational assimilation with a neural network architecture. I recommend publication after the authors address two concerns.

Line 118, 151-160: Emulation of a physical model (script M) with a neural network (Phi) is non-trivial and a significant result on its own. In similar studies, emulation of the dynamical model results in small error for each timestep but leads to very significant error growth over time. Would the authors comment on the trends in error of Phi against the nature run when there is no assimilation? Temporal error performance is generally good in this paper, suggesting assimilation is successful in preventing error growth on a two-month timescale, but the upward trend in the 4DVarNet 1 swot + 4 nadir error in Figure 9b is a possible cause of concern.

Section 5.4: The ensemble method for 4DVarNet-SSH uncertainty quantification presented here is in fact measuring the uncertainties introduced by stochastic gradient descent, not other aspects of the assimilation scheme which one would expect to contribute to the interpolation error. Correlation with the interpolation error does not necessarily imply interpretability as interpolation error, as the different scales in Figure 13 indicate. With this limitation in mind, could the authors be more precise in their comment on lines 348-9: "As such, it can be regarded as an indicator of the interpolation error."

In addition, while reviewing I came across a few typos and other minor issues that should be corrected in the revised manuscript:

Line 66: Omega argument in cost function is not defined until Algorithm 1 box; please define here.

Line 81: smoothing instead of smooting

Line 118: with instead of which in "neural operator phi which convolutional architecture".

Line 122: no script L in the equation, only LSTM. h(i) and c(i) not defined. Revisit definition of terms to make sure everything is clearly laid out for the reader.

Algorithm 1 box: no e at the end of algorithm

Line 176: This statement is unclear, revise: "Let precise that state sequences"

Some figures have very small text and are hard to read. For example, consider increasing font sizes in Figures 6, 7, and 8.

Not all elements of figure 8b and 9b are labelled in the legend or caption.

**Response to reviewer #2 :**

Dear Sir or Madam,

Happy new year and thanks a lot for your comments. I will try to answer your questions:

Line 118, 151-160: Emulation of a physical model (script M) with a neural network (Phi) is non-trivial and a significant result on its own. In similar studies, emulation of the dynamical model results in small error for each timestep but leads to very significant error growth over time. Would the authors comment on the trends in error of Phi against the nature run when there is no assimilation? Temporal error performance is generally good in this paper, suggesting assimilation is successful in preventing error growth on a two-month timescale, but the upward trend in the 4DVarNet 1 swot + 4 nadir error in Figure 9b is a possible cause of concern.

> In this work, the operator Phi is learnt jointly with the solver Gamma. In the paper, we said that we substitute to M the neural operator Phi. I prefer this formulation rather than emulation, because Phi is not actually emulating M. Indeed, it means that during the training, Phi (the prior in the variational cost J) is optimized jointly with Gamma to satisfy at most the training loss function (basically the MSE with some regularization terms). This can be seen as a possible answer to the question: "what is the best prior to use in DA?". Then, Phi should not be compared to a model M in this study, there is no physical model at all (we only use it as the ground truth in this paper). As a consequence, we cannot really study the error growth of Phi over time, because we are solving an interpolation problem. We plan to look at such questions when applying the 4DVarNet formulation on similar forecasting problems.

On the last point and consequently, I think the (small) upward trend in the 4DVarNet 1 swot + 4 nadir error of Figure 9b is simply due to the observation sampling.

Section 5.4: The ensemble method for 4DVarNet-SSH uncertainty quantification (UQ) presented here is in fact measuring the uncertainties introduced by stochastic gradient descent, not other aspects of the assimilation scheme which one would expect to contribute to the interpolation error. Correlation with the interpolation error does not necessarily imply interpretability as interpolation error, as the different scales in Figure 13 indicate. With this limitation in mind, could the authors be more precise in their comment on lines 348-9: "As such, it can be regarded as an indicator of the interpolation error."

> I completely agree with you, such an approach only provides a measure to quantify the uncertainty of the prior and solver joint training. As said l.345, it [only] reveals the internal variability of the training process. Then, it does not adress all the components of the interpolation error related to the DA scheme used here. We plan in future works to draw from traditional ensemble DA methods or ensemble gaussian-based simulations to address such a task. Regarding our comments on line 348-9, I think it has to be more explicit. I proposed

this sentence: "[...] the standard deviation of the ensemble of 4DVarNet-SSH schemes correlates to the interpolation error, with an R2 coefficient of determination equals to 0.86. Even if the scales between the interpolation error and the training related 4DVarNet internal variability differ, see Fig. 13, the latter can be regarded as an indicator of the interpolation error, usually with an appropriate localization of large errors. In future works, we plan to draw from traditional ensemble DA methods or ensemble gaussian-based simulations to address all the components of the interpolation error related to the data assimilation scheme."

In addition, while reviewing I came across a few typos and other minor issues that should be corrected in the revised manuscript:

Line 66: Omega argument in cost function is not defined until Algorithm 1 box; please define here.

> Omega is defined line 72 after the full equation of the cost function.

Line 81: smoothing instead of smooting

> Modification done

Line 118: with instead of which in "neural operator phi which convolutional architecture".

> Modification done: "based on a convolutional architecture"

Line 122: no script L in the equation, only LSTM. h(i) and c(i) not defined. Revisit definition of terms to make sure everything is clearly laid out for the reader. > Modifications and precisions made in the new version of the manuscript

Algorithm 1 box: no e at the end of algorithm

> Modification done

Line 176: This statement is unclear, revise: "Let precise that state sequences"

> Reformulation: "Let precise that we use a data assimilation window of length $N=2l+1$, meaning that we use temporal sequences of state space ..."

Some figures have very small text and are hard to read. For example, consider increasing font sizes in Figures 6, 7, and 8.

> I increased the font sizes

Not all elements of figure 8b and 9b are labelled in the legend or caption.

> I think you mentioned the red and green colorbars. I precised in the caption what they represent : "the daily spatial coverage of the two configurations are given as complementary red and green barplots scaled on the right-hand side"

Kind regards,